# Feature-specific reaction times reveal a semanticisation of memories over time and with repeated remembering

Julia Lifanov [1✉], Juan Linde-Domingo [2] & Maria Wimber [1,3✉]

Memories are thought to undergo an episodic-to-semantic transformation in the course of their consolidation. We here test if repeated recall induces a similar semanticisation, and if the resulting qualitative changes in memories can be measured using simple feature-specific reaction time probes. Participants studied associations between verbs and object images, and then repeatedly recalled the objects when cued with the verb, immediately and after a two-day delay. Reaction times during immediate recall demonstrate that conceptual features are accessed faster than perceptual features. Consistent with a semanticisation process, this perceptual-conceptual gap significantly increases across the delay. A significantly smaller perceptual-conceptual gap is found in the delayed recall data of a control group who repeatedly studied the verb-object pairings on the first day, instead of actively recalling them. Our findings suggest that wake recall and offline consolidation interact to transform memories over time, strengthening meaningful semantic information over perceptual detail.

[1] School of Psychology and Centre for Human Brain Health (CHBH), University of Birmingham, Edgbaston, Birmingham, UK. [2] Center for Adaptive Rationality, Max Planck Institute for Human Development, Berlin, Germany. [3] Institute of Neuroscience & Psychology and Centre for Cognitive Neuroimaging (CCNi), University of Glasgow, Glasgow, UK. ✉email: j.lifanov.1@pgr.bham.ac.uk; maria.wimber@glasgow.ac.uk

One powerful way to protect memories against forgetting is to recall them frequently. Decades of research on the testing effect have shown such a protective effect, suggesting that repeated remembering stabilizes newly acquired information in memory[1–6]. It is unknown, however, whether all aspects of memory equally benefit from active recall. The aim of the present work was to investigate the qualitative changes in memories that occur with time and repeated remembering. We used feature-specific reaction time (RT) probes to measure such changes in lab-based visual memories. Specifically, we expected to observe a transformation along a detailed-episodic to gist-like-semantic gradient, based on several strands of research indicating that memories become "semanticised" in the process of their stabilisation.

Dominant theories of the testing effect make the central assumption that active recall engages conceptual-associative networks more so than other practice techniques such as repeated study[7–9]. The elaborative retrieval account suggests that during recall, a conceptual relationship is established between initially separate episodic elements to unify them into a coherent memory[9]. Similarly, the mediator effectiveness hypothesis[10] states that testing promotes long-term retention by evoking mediator representations, which are concepts that have meaningful overlap with a memory cue and target[7]. Together, this work suggests that remembering co-activates semantically related concepts, more than restudy, and can thereby contribute to the long-term storage of newly acquired memories by linking them to already established, related concepts.

Other authors have made similar assumptions from a more neurobiologically and computationally motivated perspective[11], drawing a parallel between the processes stabilizing memories via online recall, and the processes thought to consolidate memories via offline replay, including during sleep. In this online consolidation framework of the testing effect, active recall activates a memory's associative index in the hippocampus, together with the neocortical nodes representing the various elements contained in the memory. As a result of this simultaneous activation, links between the active elements are strengthened[12]. Moreover, because recall tends to be somewhat imprecise, more so than re-encoding, activation spreads to associatively or conceptually related elements, providing an opportunity to integrate the new memory with related information. This presumed stabilization and integration is strongly reminiscent of the hippocampal-neocortical dialogue assumed to happen during sleep-dependent memory replay[13], resulting in the integration of new memories into existing relational knowledge, and the strengthening of conceptual/schematic links between memories[14]. Critically, many consolidation theories assume that this reorganization goes along with a "semanticisation" of memories, such that initially detail-rich episodic memories become more gist-like and lose detailed representations over time and with prolonged periods of consolidation[15–18]. Based on these parallels between wake retrieval and offline consolidation, the present study tested whether repeated recall specifically induces a behaviourally measurable "semanticisation" that goes beyond the effects that naturally occur over time.

In the human memory consolidation literature, much of the empirical evidence for semanticisation comes from neuroimaging studies showing a gradual shift in the engagement of hippocampus and neocortex during recent and remote recall, or studies tracking representational changes in memories over time[16,19]. Recent work even suggests that the neocortical changes that accompany such shifts can occur rapidly, across repeated exposures to episodic events on the same day[20,21], and that these changes are then further stabilized through subsequent periods of sleep[22]. Behavioural studies, on the other hand, have largely relied on scoring of autobiographical or other descriptive verbal memory reports for central gist versus peripheral details, and yielded robust evidence for a detail-to-gist gradient[23,24]. The present study used a different approach, asking if semanticisation via recall can be observed in RTs that specifically reflect the speed with which participants can access higher-level conceptual and lower-level perceptual features of visual object memories.

This method was recently introduced by Linde-Domingo et al.[25]. They showed that when participants are retrieving visual objects from memory, conceptual aspects (e.g., Does the recalled image represent an animate or inanimate object?) are accessed more rapidly than perceptual aspects (e.g., Does the recalled image represent a photo or a drawing?). In sharp contrast, RTs were consistently faster to perceptual than conceptual questions when the image was physically presented on the screen. This flip suggests that recalling a memory progresses in the opposite direction from visual perception, reactivating the core meaning first before back-propagating to sensory details. Such semantic prioritisation is plausible considering that the hippocampus is most directly and reciprocally connected with late sensory processing areas assumed to represent abstract concepts[26,27]. Both online retrieval and offline replay of hippocampus-dependent memories can therefore be assumed to preferentially activate conceptual features of a memory, and this prioritisation may over time produce a semanticised memory compared with the one originally encoded. With this background in mind, and an adapted version of the described RT paradigm, we here investigate whether repeated retrieval enhances the semanticisation of memories over time compared to repeated study.

In this work, two groups of participants learn novel verb-object pairings at the beginning of day 1 (Fig. 1), and then immediately practice the associations six times overall. Subjects in the retrieval group practice by actively recalling the object image from memory when cued with the verb, and answering conceptual and perceptual questions about the recalled object as fast as possible. Subjects in the restudy group instead practice by re-encoding the intact verb-object pairings, answering the conceptual or perceptual question while seeing the object on the screen. All participants return to the lab 48 h later for a delayed cued recall test, where each verb-object pairing is probed once more with a conceptual and a perceptual question. Feature-specific RTs are used as a measure of accessibility to lower-level perceptual or higher-level conceptual object features. We show that the RT gap between perceptual and conceptual features increases across the two-day delay, indicative of time-dependent semanticisation, and that active retrieval plays a central role in this presumed semanticisation. Dependency analyses also suggest that object features are remembered and forgotten in a hierarchical fashion, and that recall becomes more dependent on conceptual object features over time.

## Results

**Semanticisation over time**. Participants in the retrieval group ($n = 49$) immediately practiced the newly learned verb-object associations via cued recall. They did so six times overall, in three pseudo-randomized cycles that each contained one perceptual and one conceptual feature probe. We first tested the retrieval group data for a time-dependent semanticisation, assuming that memory recall prioritises access to conceptual over perceptual features, and that this prioritisation increases over the two days with increasing semanticisation. Importantly, we wanted to isolate the transformation of episodic memories that occurs purely with passage of time, as opposed to the changes that occur already on the first day across the repeated practice trials. Using feature-specific RTs, we thus compared the memory representation at the end of day 1 (i.e., cycle 3), after completed learning and practice, to the representation on day 2. We expected an increased perceptual–conceptual RT gap on the delayed

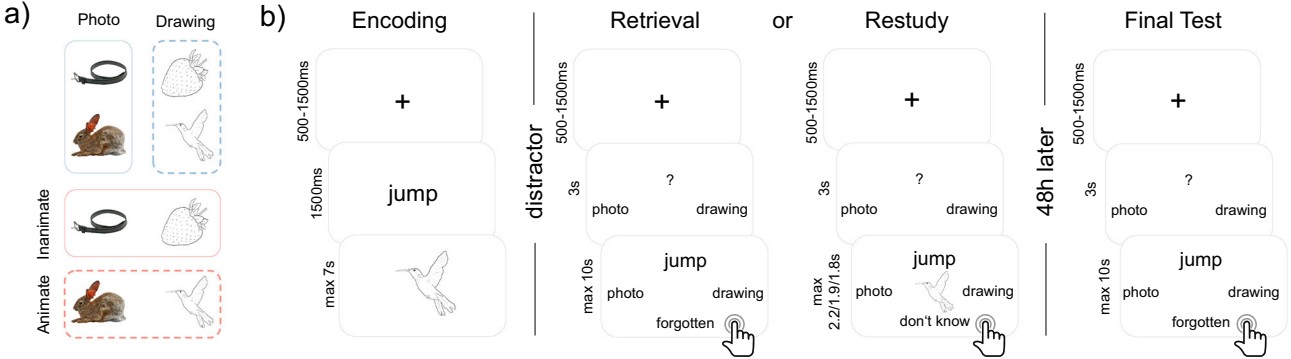

**Fig. 1 Overview of stimuli and task. a** Design of the stimuli. The 64 pictures used in any given participant were orthogonally split into 32 drawings and 32 photographs, out of which 16 were animate and 16 inanimate objects, respectively. Each object could thus be classified along a perceptual (photo/drawing, blue) or conceptual (animate-inanimate, red) dimension. **b** One prototypical task block of the paradigm within the repeated retrieval/restudy group. Both groups performed eight blocks, each starting with the encoding of eight novel verb-object associations. After a 20 s distractor task, each of the eight associations was practiced twice in each of the three practice cycles, once with a conceptual, once with a perceptual question, and reaction times (RTs) were measured on each of the overall six practice trials. The maximum response time in each practice cycle of the restudy group was set to the average response time of the corresponding cycle in the retrieval group. After 48 h, participants returned to complete a final test, where again each association was tested once with each of the two question types, with RTs being recorded, as indicated by the button press symbols. Finally, a written cued recall test was performed. Stimuli depicted are chosen from the BOSS database (https://sites.google.com/site/bosstimuli/home [59], https://creativecommons.org/licenses/by-sa/3.0/) and customized with free and open source GNU image manipulation software (www.gimp.org; see Linde-Domingo et al.[25]).

day 2 test, compared to the end of day 1. A 2 (recall cycle: end of day 1 vs day 2) by 2 (question type: conceptual vs perceptual) repeated measures analysis of variances (rmANOVA) on the RT data of the repeated retrieval group only (Fig. 2) showed a main effect of recall cycle ($F(1,48) = 71.44$, $p < 0.01$) indicating slower responses on day 2 than day 1, and the main effect of question type ($F(1,48) = 29.58$, $p < 0.01$) with conceptual questions being consistently answered faster than perceptual questions. Critical to our first main hypothesis, the rmANOVA also revealed a significant interaction ($F(1,48) = 19.87$, $p < 0.01$) between the two factors, indicating that the conceptual-over-perceptual RT advantage changed across days. A posthoc power analysis in G*Power revealed an effect size of $d = 0.64$ and a power of 0.99 for the interaction effect. Average RTs confirmed that the interaction was produced by an increasing perceptual–conceptual RT gap from the end of day 1 ($M_{day1} = 40$ ms, $SD_{day1} = 194$ ms) to day 2 ($M_{day2} = 290$ ms, $SD_{day2} = 359$ ms), in line with the semanticisation hypothesis. Note that the interaction is equally robust when using the averaged day 1 RTs within participants ($F(1,48) = 20.11$, $p < 0.01$), rather than the cycle 3 data. Together, these results suggest that semantic features preferentially benefit from the passage of time after retrieval practice, in line with semanticisation.

We additionally tested whether the perceptual–conceptual gap in the retrieval group already changed across cycles on day 1, in line with a "fast consolidation" process[11]. A 3 (cycle: 1, 2, 3) by 2 (question type: conceptual or perceptual) repeated measures ANOVA of the day 1 RTs (Fig. 2) revealed a significant main effect of cycle ($F(2,96)) = 102.44$, $p < 0.01$), with participants becoming faster over time, as well as a significant main effect of question type ($F(1,48) = 5.01$, $p = 0.03$), with conceptual questions being answered overall faster than perceptual ones, generally replicating the results of Linde-Domingo et al. (2019). However, the cycle by question type interaction was not significant ($F(2,96) = 0.42$, $p = 0.66$), indicating that the perceptual–conceptual gap did not change significantly across practice cycles. The immediate recall data of this study thus provide no behavioural evidence for a fast semanticisation.

**Stronger semanticisation after repeated retrieval than restudy.** To test our second hypothesis, that repeated retrieval leads to a stronger delayed perceptual–conceptual gap than repeated study, we investigated the RT gap on the second testing day in both groups. If semanticisation over time is enhanced by retrieval practice, this should be reflected in a larger RT gap in the retrieval group ($n = 49$) in contrast to the restudy group ($n = 24$). A 2 (practice condition: retrieval vs restudy) by 2 (question type: conceptual vs perceptual) mixed ANOVA on the RTs of day 2 (Fig. 3) revealed no main effect of practice condition ($F(1, 71) = 1.41$; $p = 0.24$), and a main effect of question type ($F(1, 71) = 16.92$; $p < 0.01$) with overall shorter RTs for conceptual than perceptual questions. As hypothesized, a significant interaction was found between question type and practice condition ($F(1, 71) = 5.21$; $p = 0.03$). Our posthoc power analysis on the interaction effect revealed an effect size of $d = 0.27$ and a power of 0.99. This interaction was due to an effect in the expected direction, with a higher perceptual–conceptual difference in the repeated retrieval group ($M_{retrieval} = 290$ ms, $SD_{retrieval} = 359$ ms) than in the restudy group ($M_{restudy} = 83$ ms, $SD_{restudy} = 372$ ms), in line with the interpretation that repeated retrieval leads to more pronounced semanticisation than repeated study. Specifically, we found that perceptual questions are answered only slightly faster in the retrieval group ($M_{retrieval} = 2.63$ s, $SD_{retrieval} = 0.75$ s) in comparison to the restudy group ($M_{restudy} = 2.74$ s, $SD_{restudy} = 0.86$ s). In contrast, conceptual RTs show a comparatively stronger difference between repeated retrieval ($M_{retrieval} = 2.34$ s, $SD_{retrieval} = 0.61$ s) and restudy ($M_{restudy} = 2.66$ s, $SD_{restudy} = 0.84$ s), suggesting that the interaction is mainly caused by faster access to conceptual features in the retrieval group, in line with a retrieval-induced semanticisation. A subsampling analysis confirmed that the differential RT gap is also robustly found when equating the sample sizes of the retrieval and the restudy group (see Supplementary Methods and Supplementary Fig. 1).

**A replication of the reversed retrieval stream.** Next, we analysed the data of the first day to test if we could replicate a reversal of the RT patterns between memory retrieval and visual exposure, conceptually replicating the previous results[25]. Based on these findings, we expected faster RTs to conceptual than perceptual questions (i.e. a reverse stream) in the retrieval group that practiced the associations via active recall (Fig. 2), and faster perceptual than conceptual RTs (i.e., a forward stream) in the restudy group that practiced the associations by visual re-exposure. We therefore performed a mixed 2 (practice condition: retrieval vs restudy) by 2 (question type:

## Repeated retrieval RTs to each question

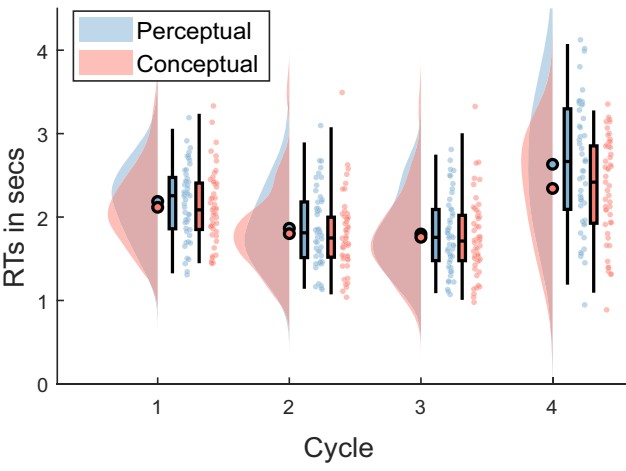

**Fig. 2 Retrieval reaction times (RTs), separate by question and repetition.** The gap between perceptual and conceptual RTs significantly increases from day 1 to day 2 (interaction between question type and repetition ($F(1,48) = 19.87$, $p = 0.00$, $d = 0.64$)), in line with a semanticisation process over time. RTs in the repeated retrieval group are depicted for each cycle and question type. Filled circles represent the overall mean, boxplots represent median and 25th and 75th percentiles; whiskers represent 2nd and 98th percentile; dots represent the means of individual subjects. Blue represents perceptual, red conceptual responses. $N = 49$ independent subjects. Source data are provided as a Source Data file.

## Perceptual-conceptual gap in each group

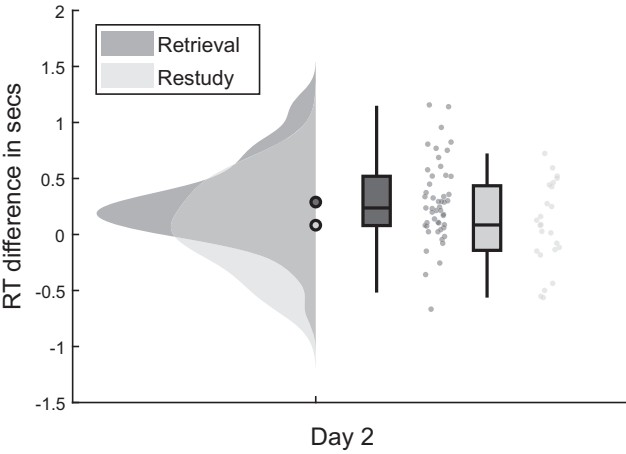

**Fig. 3 Perceptual-conceptual reaction time (RT) gap in each group on day 2.** Repeated retrieval yields a stronger perceptual–conceptual RT gap than restudy on the delayed test (two-sided $t(71) = 2.28$, $p = 0.03$, CI = [0.03, 0.39]). Perceptual-conceptual RT gaps are depicted for retrieval and restudy on day 2. Filled circles represent the overall mean, boxplots represent median and 25th and 75th percentiles; whiskers represent 2nd and 98th percentile; dots represent the means of individual subjects. Dark grey represents retrieval data, light grey represents restudy data. $N = 49$ independent subjects in the retrieval group, $n = 24$ independent subjects in the restudy group. Source data are provided as a Source Data file.

conceptual vs perceptual) ANOVA on the day 1 data, averaging RTs across the 3 cycles. Apart from a main effect of task ($F(1,71) = 71.13$, $p < 0.01$), and no main effect of question type ($F(1,71) < 0.01$, $p = 0.98$), this analysis revealed the expected, significant cross-over

interaction ($F(1,71) = 9.24$, $p < 0.01$) with faster responses for perceptual questions than conceptual ones in restudy ($M_{per} = 1.13$ s, $SD_{per} = 0.21$ s; $M_{con} = 1.19$ s, $SD_{con} = 0.19$ s) and vice versa in retrieval ($M_{per} = 1.95$ s, $SD_{per} = 0.42$ s; $M_{con} = 1.89$ s, $SD_{con} = 0.44$ s).

**Hierarchical relationship between remembered features.** Two further analyses were conducted on accuracy data, rather than RTs. First, we investigated a possible hierarchical dependency between perceptual and conceptual features as shown in recent work[28] and how this relationship changed over time. All correct and incorrect recall trials with a response time above 200 ms were sorted into four categories, depending on whether participants remembered both features, only perceptual features, only conceptual features or none. In line with previous work[28,29], we expected that over time, the majority of items would be forgotten in a holistic manner, such that items that were fully remembered ("both features correct") on day 1 would be fully forgotten ("none correct") on day 2. For the present purpose, we were however particularly interested in the two response categories indicating partial remembering (i.e., "conceptual only" and "perceptual only" recall trials). Here, a hierarchical dependence in a reverse memory reconstruction stream predicts a particular pattern: higher-level conceptual information would need to be accessed before the lower-level perceptual information can be reached. As a result, participants should be relatively likely to remember the conceptual feature ("Was it animate or inanimate") while forgetting the perceptual one ("Was it a photo or drawing"), but there should be very few trials where they remember the perceptual while forgetting the conceptual feature, except for random guesses. We thus expected to see a significant difference in the number of responses falling into these two categories already on the immediate day 1 recall. If semanticisation increases this hierarchical dependency, the gap in the proportion of conceptual-only and perceptual-only recalls should significantly increase across the 2-day delay.

We carried out a 2 (recall cycle: end of day 1 vs day 2) by 2 (features remembered: conceptual-only vs perceptual-only) rmANOVA to test this hypothesis. This analysis revealed a main effect of repetition ($F(1,48) = 53.97$, $p < 0.01$), and a main effect of features remembered ($F(1,48) = 27.10$, $p < 0.01$), the latter effect in line with hierarchical recall. Importantly, we also found the expected significant interaction ($F(1,48) = 8.21$, $p < 0.01$), reflecting the observation that over time, the number of objects for which the conceptual but not the perceptual feature could be remembered increased significantly more than the number of objects for which the opposite pattern was true (Fig. 4). The interaction is equally robust when using the averaged day 1 accuracies ($F(1,48) = 14.31$, $p < 0.01$), rather than the cycle 3 data.

Note that the data presented in Fig. 4 is not corrected for estimated random guesses[28], as such a correction would have turned most proportions negative, and therefore seemed to be an overestimation of guesses in our dataset. However, since the guesses of a particular cycle are assumed to be distributed equally across response categories within that cycle, correcting does not change the outcomes of the statistical analysis (corrected repetition effect $F(1,48) = 55.52$, $p < 0.01$), corrected features remembered effect ($F(1,48) = 27.10$, $p < 0.01$), and corrected interaction ($F(1,48) = 8.21$, $p < 0.01$)). Again, the interaction was also significant when comparing the average day 1 data to day 2 ($F(1,48) = 14.31$, $p < 0.01$).

**A replication of the testing effect.** Finally, we also assessed the written cued recall responses on the second day to investigate if a general testing effect was found in our sample. To do so, we compared the accuracy in the written sheet responses between both

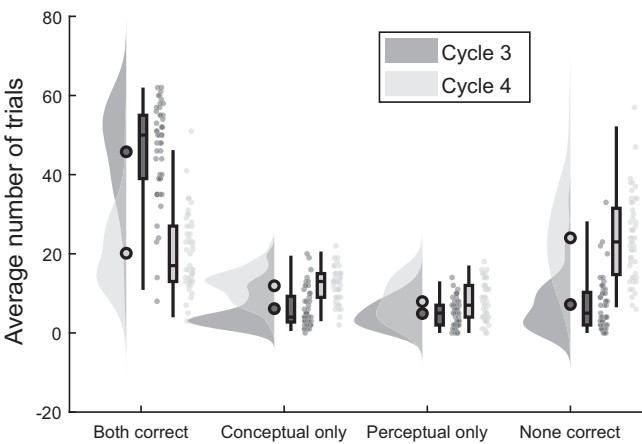

## Remembered object features per cycle

**Fig. 4 Remembered and forgotten object features, separate for cycles 3 and 4.** The data show a hierarchical dependency between conceptual and perceptual memory features that increases over time (interaction between feature remembered and cycle $F(1,48) = 8.21$, $p = 0.01$, $d = 0.41$). The average number of objects in each response-category for cycle 3 (end of day 1) and cycle 4 (day 2) of all subjects in the repeated retrieval group are shown. Filled circles represent the overall mean, boxplots represent median and 25th and 75th percentiles; whiskers represent 2nd and 98th percentile; dots represent the means of individual subjects. Dark grey represents cycle 3, light grey represents cycle 4. $N = 49$ independent subjects. Source data are provided as a Source Data file.

experimental groups, using two independent sample $t$-tests. All written responses were categorized by two experimenters as "specific correct/incorrect" and "coarse correct/incorrect" responses. Here, specific correct includes retrieving the exact object label (e.g., parrot), whereas coarse correct responses also include correct descriptions of the object's category (such as "colourful bird" for "parrot"). Two-sided $t$-tests revealed that participants in the repeated retrieval group ($M_{coarse} = 0.30$, $SD_{coarse} = 0.19$; $M_{specific} = 0.25$, $SD_{specific} = 0.18$) recalled significantly more associations than restudy participants ($M_{coarse} = 0.20$, $SD_{coarse} = 0.18$; $M_{specific} = 0.16$, $SD_{specific} = 0.17$) using either scoring scheme, specific ($t(71) = 2.06$, $p = 0.04$, CI = [0.00, 0.17]) and coarse ($t(71) = 2.16$, $p = 0.03$, CI = [0.01, 0.19]; see Supplementary Methods and Supplementary Fig. 1).

We were then interested whether the paper-and-pencil based cued recall performance was related to the size of the perceptual–conceptual gap in feature-specific RT probes. We thus tested whether, within participants, those items for which delayed memory performance is strong enough to support cued recall on the final sheet test show a larger RT gap. To do so, all RTs of day 2 were compared between trials where the corresponding object was remembered on the response sheets (specific/coarse correct), and those where the object was not remembered (specific/coarse incorrect). Results show that the perceptual–conceptual RT gap is significantly larger for correctly than for incorrectly recalled items ($t(71) = 2.13$, $p = 0.04$, CI = [0.01, 0.33] with the specific scoring approach, $t(71) = 2.65$, $p = 0.01$, CI = [0.05, 0.38] with the coarse scoring approach). Decomposing these differences in more detail, we found similar perceptual RTs for correct and incorrect sheet responses (for coarse scoring: $M_{per\_corr} = 2.54$ s, $SD_{per\_corr} = 0.87$ s, $M_{per\_incorr} = 2.52$ s, $SD_{per\_incorr} = 0.97$ s; for specific scoring: $M_{per\_corr} = 2.50$ s, $SD_{per\_corr} = 0.68$ s, $M_{per\_incorr} = 2.52$ s, $SD_{per\_incorr} = 0.97$ s) whereas the conceptual RTs for correct sheet responses (coarse scoring: $M_{con\_corr} = 2.23$ s, $SD_{con\_corr} = 0.59$ s; specific scoring: $M_{con\_corr} = 2.20$ s, $SD_{con\_corr} = 0.62$ s)

are faster than those for incorrect sheet responses (coarse scoring: $M_{con\_incorr} = 2.42$ s, $SD_{con\_incorr} = 0.93$ s; specific scoring: $M_{con\_incorr} = 2.40$ s, $SD_{con\_incorr} = 0.89$ s). In contrast to the difference between correct and incorrect perceptual RTs ($t(71) = 0.03$, $p = 0.98$, CI = [−0.19, 0.19]), the difference between conceptual RTs for correct and incorrect sheet responses is significant with the coarse scoring method ($t(71) = −2.49$, $p = 0.02$, CI = [−0.38, −0.04]) and thus seems to drive the changed RT gap on day 2 (the specific scoring method did not yield a significant difference for neither feature; $t(71) = −0.22$, $p = 0.83$, CI = [−0.21, 0.17] for perceptual; $t(71) = −1.95$, $p = 0.06$, CI = [−0.39, 0.00] for conceptual questions; Fig. 5). Again, these findings suggest that items for which a strong episodic trace exists show a larger RT gap, caused by relatively faster access to conceptual item features. Note that we found a mirrored effect for accuracies, such that participants who performed better on the paper-and-pencil test showed a larger accuracy gap in the button presses, when splitting according to specific ($t(71) = 3.08$, $p < 0.01$, CI = [0.03, 0.15]) and the coarse ($t(71) = 3.95$, $p < 0.01$, CI = [0.05, 0.16]) scoring method (Supplementary Fig. 2).

## Discussion

Do memories change every time we remember them? Cognitive[10,30] and neurobiologically motivated[11] theories assume that each active recall constitutes a distinct online consolidation event that systematically changes the nature of the memory, from an initially detail-rich episode to a "semanticised" version of the same event. Two questions were of central interest in the present study. First, we wanted to test if feature-specific probes can be used to reveal this presumed perceptual-to-conceptual transformation (semanticisation) of memories over an initial period of consolidation. Second, we were interested if repeated remembering specifically boosts this transformation compared with repeated study, preserving conceptual information relatively more over time.

To test our first hypothesis of a semanticisation over time, we measured how fast participants were able to recall perceptual and conceptual features of previously memorised objects on the first day, compared with how fast they accessed the same features 48 h later. While conceptual information was consistently accessed faster on the immediate and the delayed memory test, the perceptual–conceptual gap significantly increased over the two-day retention period, suggesting that access to conceptual memory features was favoured over access to perceptual features across the temporal delay. This finding is consistent with at least two possible interpretations. High-level semantic information may be prioritised for active consolidation, an ongoing discussion in the consolidation literature[16,31]. Or semantic information might be forgotten at a slower rate than perceptual information, a possibility we return to further below. As also elaborated below, hierarchical forgetting and prioritisation for active consolidation may in fact rely on the same underlying mechanism.

Recent studies do support an active and selective consolidation view. For example, structured, categorical information shows above-baseline enhancement from sleep, compared with detailed, stimulus-unique features of the memorized stimuli[32]. It has thus been suggested that structured information is subject to active consolidation. In terms of functional anatomy, the hippocampus is most directly connected with late sensory areas coding abstract-semantic features of objects[26,27]. Moreover, concept cells in the hippocampus are thought to form the building blocks of episodic memories[33]. The elements (e.g., objects, people) that constitute an episode are thus likely bound together on the level of meaningful semantic units. During retrieval and offline replay, it is assumed that the linked elements belonging to the same episode are

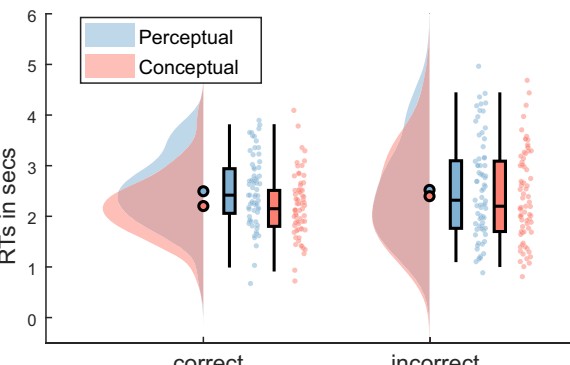

a) Perceptual and conceptual RTs for (in)correct specific sheet responses day 2

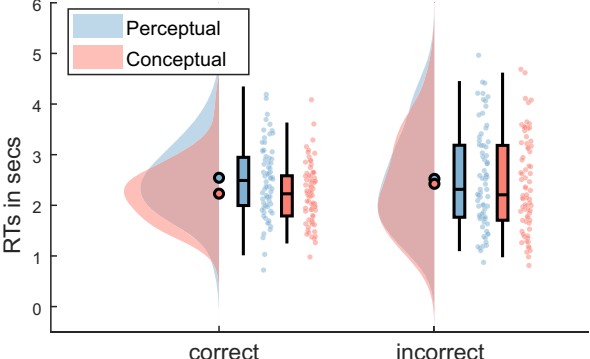

b) Perceptual and conceptual RTs for (in)correct coarse sheet responses day 2

**Fig. 5 Perceptual and conceptual reaction times (RTs) for (in)correct specific and coarse written cued recall responses on day 2.** Only associations that are remembered in the cued recall test yield a perceptual–conceptual reaction time gap in the task (two-sided test for the gap difference between correctly and incorrectly remembered associations: $t(71) = 2.13$, $p = 0.04$, $CI = [0.01, 0.33]$ with the specific scoring approach, two-sided $t(71) = 2.65$, $p = 0.01$, $CI = [0.05, 0.38]$ with the coarse scoring approach). Perceptual and conceptual RTs are shown as categorized in correct and incorrect specific (**a**)/coarse (**b**) sheet responses. Filled circles represent the overall mean, boxplots represent median and 25th and 75th percentiles; whiskers represent 2nd and 98th percentile; dots represent the means of individual subjects. Blue represents perceptual, red conceptual responses. $N = 73$ independent subjects. Source data are provided as a Source Data file.

reactivated in a cascade that starts with pattern completion in the hippocampus, followed by a back-propagation into neocortex[34–36]. This back-propagation likely starts off with the information coded closest to the hippocampus, and then progresses backwards along the neocortical hierarchy[25]. The presumed hippocampal-neocortical dialogue during wake retrieval and sleep might thus prioritise conceptual features of the reactivated memories, relative to their perceptual features that are coded in brain areas further removed from the hippocampus. As a result, each replay event would strengthen semantic information more than perceptual, further exaggerating the gap that is already present on the immediate recall.

Alternatively, it is possible that the perceptual features of our visual objects were forgotten faster than their conceptual features. The nature of item-based forgetting is still under debate[37]. Some recent work suggests that the forgetting of perceptual features, such as colour, is independent of, and occurs faster than, forgetting of higher-level conceptual features such as item state or exemplar[38,39]. In contrast, other research has shown that object memories are forgotten in a more holistic manner, with an interesting hierarchically dependent forgetting of perceptual and conceptual features[28]. Inspired by this work, we investigated a possible hierarchical dependency of forgetting in our own accuracy patterns. We indeed saw evidence for asymmetrical recall, such that if participants only recalled one of the two features, they were more likely to remember the conceptual but not the perceptual feature than vice versa. This asymmetry significantly increased over the two-day delay, again indicating an increasing dependence of remembering on conceptual features. Together with the RT results, our findings therefore support a view of hierarchically dependent remembering and forgetting of single item features, with lower-level perceptual features having a higher likelihood of being forgotten independently of higher-level semantic features.

To distinguish the contribution of active retrieval to such time-dependent consolidation effects, we further tested whether retrieval on the first day enhances the preservation of conceptual features more than restudy, a more visual type of practice that does not involve the same degree of intrinsic memory

reactivation. In line with our second main hypothesis, we found a larger perceptual–conceptual RT gap on the second day in the retrieval group. Average RTs indicate that this change is driven by a pronounced gain in conceptual feature access after repeated retrieval compared to restudy, rather than differences in how fast the two groups access the perceptual details of the stimuli. This pattern of results suggests that the underlying semanticisation process is relatively stronger when the originally learned associations are immediately practiced by active cued recall, and it has at least two important implications.

First, our finding has implications for theories of the testing effect, showing that active recall disproportionally increases the access to conceptual aspects of memory over perceptual aspects. This finding resonates with the idea that each memory recall tends to co-activate semantically related information[40], in turn facilitating the integration of newly learned information into existing knowledge networks[10,30]. In the long-term, such knowledge integration can then aid memory recall, as supported by the close relationship we found between paper-and-pencil accuracy and the RT gap on day two, such that participants with good episodic recall tend to show larger differences between perceptual and conceptual feature access on the delayed tests. Semanticisation thus seems to support episodic recall on delayed tests and is boosted by retrieval practice.

Second, our results suggest that repeated remembering could be an important factor for representational memory changes that interact with a subsequent period of sleep (see also Cairney et al.[41]). Many sleep studies carry out a memory test before and after sleep to obtain a difference score within subjects[42], and it is thus important to distinguish retrieval's specific contribution to the observed consolidation effects. While the present study suggests that repeated recall can amplify time-dependent, qualitative changes in memories, we did not manipulate whether the retrieval was followed by a period of wake or sleep, and our results can therefore not directly address the interaction between retrieval practice and sleep. One recent study found that sleep's benefits were indeed reduced when preceded by retrieval compared to restudy practice[43]. These results, however, were interpreted as evidence that repeated recall can strengthen memories

up to a point where they no longer benefit from sleep[43], an interpretation that is well in line with other reports that sleep prioritizes, or at least has a more measurable effect on, weaker memories[44,45]. In such cases, more sensitive tests like our feature-specific RT probes, or test conditions with higher demands (e.g. on interference control) can still be used to successfully uncover consolidation effects[44]. In future studies, sensitivity and the timing of tests will certainly play an important role in disentangling the differential contributions of repeated remembering and sleep to the long-term retention of memories.

Our findings support the idea that active testing, in terms of the neurobiological processes involved, mimics consolidation by relaying newly acquired information from hippocampal to neocortical structures[11]. However, the perceptual–conceptual gap in the retrieval group did not change with repeated remembering on day 1, and our results do thus not provide evidence for a "fast" consolidation[11] process. A very similar finding was recently reported in a study investigating qualitative changes in memory-specific, multivariate neural patterns during recall[46]. In contrast, recent neuroimaging studies report that functional and neuroplastic changes can occur within a single session of rehearsal[20,21], speaking in favour of a fast consolidation process. Why then do the behavioural, feature-specific changes not become visible on a more rapid timescale, when the neurobiological changes presumably take place? After initial learning and practice, hippocampal and neocortical memory traces likely co-exist, with the hippocampus dominating immediate recall. Delayed recall has been shown to depend more strongly on neocortex[42,46,47], in line with most consolidation theories[17,48–50]. Moreover, imaging work suggests that the rapid neurophysiological markers of plasticity may require a large number of, and more spaced, repetitions to evolve[20,21], plus a further period of consolidation to become stabilised[22]. Retrieval practice might thus help to establish a neocortical trace rapidly, more so than restudy, but this trace will only become behaviourally relevant at longer delays, when remembering is more dependent on neocortex.

The present findings suggest that RTs, paired with questions that differentially probe access to specific mnemonic features, are sensitive to the presumed time- and recall-dependent transformation of relatively simple, visual-associative memories. Our feature-specific RT method thus lends itself well to testing for qualitative changes of practice and sleep on memory, and it complements other approaches that are commonly used. These include the scoring of autobiographical memories according to how much gist or detailed information subjects report (e.g. used in recent work[23]); recognition-based measures using familiarity as a proxy for gist, and recollection as a proxy for detail[51]; and more recently, measures of access and precision[52,53]. RTs are rarely used in memory studies. Object recognition work, however, shows that the speed with which participants can categorize objects (e.g., animate/inanimate) is well aligned with the time points when the same categories can be decoded from brain activity[54,55], and a recent study tracked the back-propagation of information during memory recall using such feature-specific RTs[25]. The present results indicate that differential RTs can directly tap into the qualitative changes that occur over the course of memory consolidation.

We have framed these changes as reflecting a transition along a perceptual-to-conceptual gradient, whereas the primary terminology in the consolidation literature is that of an episodic-to-semantic transition[56,57]. These gradients are clearly overlapping in our paradigm, where answering the perceptual questions requires more vivid and detailed recollection than answering the conceptual questions. Moreover, semantic features (e.g. that a dog is animate) are inherent in an object's identity, whereas our perceptual features are random bindings, and retrieving them

should thus strongly engage episodic-associative memory processes. Having said that, if our RT task primarily measured an episodic-semantic distinction, we would expect to see that participants with good episodic memory show very fast RTs to perceptual questions, diminishing the RT gap to conceptual questions. Contrary to this prediction, we found that better memory accuracy (based on the paper-and-pencil cued recall test) was related to a larger RT gap, and to faster RTs for conceptual features in particular. Access to the episodic trace does therefore not seem to scale with access to the perceptual features, even though the two processes are certainly not independent. We instead argue that feature-specific RTs reflect the neocortical back-propagation process that follows initial access to the episodic trace[25].

In summary, using feature-specific probes, we provide evidence for the semanticisation of memories over time and specifically with repeated remembering. Our main results are consistent with a framework where the natural prioritisation of conceptual information during repeated retrieval[25] has a lasting effect on what is being retained over time. We reconcile cognitive theories of the testing effect with neurobiologically motivated theories of memory retrieval, which posit that functional anatomy during retrieval dictates faster access to later, more abstract-conceptual stages of visual processing. Finally, our feature-specific RT probes provide a simple way to assess the qualitative changes of mnemonic representations over time, and might thus be useful for future consolidation studies using lab-based rather than autobiographical memories.

## Methods

**Participants and a priori power calculations**. Previously published work has found an effect size of $d = 0.55$ for the perceptual–conceptual gap in RTs during retrieval[25]. We expected an effect size at least as large on day 2 in the repeated retrieval group. A power analysis in G*Power[58] with $d = 0.55$, $\alpha = 0.05$ and a power of 0.9 suggested that a sample size of at least 30 was required to detect an existing effect in the retrieval group. The effect of most interest in the retrieval group was a significant interaction between testing day and question type, specifically such that the gap between conceptual and perceptual RTs would significantly increase from day 1 to day 2. The power for this interaction contrast could not be estimated a priori from the work of Linde-Domingo et al.[25]. To have sufficient power to detect an increase in the perceptual–conceptual gap, we decided to double their sample size, aiming for 48 subjects in the retrieval group (see results section for corresponding posthoc power analyses).

The second comparison of interest in this study was a contrast between the perceptual–conceptual gap on day 2 (i.e., delayed test) in the retrieval and the restudy groups. Again, since the effect size could not be estimated directly from previous work, we aimed for $n = 24$ participants in the restudy group based on (Linde-Domingo et al.[25]) using $n = 24$ within multiple groups to do between group comparisons. We thus aimed for a sample size of $n = 72$ overall for the critical comparison of the retrieval and the restudy group. Posthoc power analyses can be found in the results section.

Fifty-seven healthy volunteers from the local student population in Birmingham participated in the retrieval condition (45 female and 12 male, mean age [$M_{age}$] = 19.95, standard deviation [$SD_{age}$] = 0.79), of which eight were excluded due to absence on the second testing day or missing data. Another 26 volunteers participated in the restudy group (21 female and 5 male, $M_{age}$ = 18.92, $SD_{age}$ = 0.89), of which two were excluded due to absence on the second testing day. Our final sample thus consisted of 49 participants in the retrieval group and another 24 participants in the restudy group. All participants were informed about the experimental procedure, underwent a screening questionnaire (including sleep and consumption behaviour 24 h before the experiment) and gave their written informed consent. The research was approved by the STEM ethics committee of the University of Birmingham.

**Material**. The paradigm was an adapted version of the visual verb-object association task designed by Linde-Domingo et al.[25]. Our stimulus materials consisted of 64 action verbs and 128 pictures of everyday objects, all presented on white backgrounds (see Fig. 1.a and the previous work[25] for more detailed information about the source and manipulation of pictures[25] (BOSS database, www.gimp.org [59]) and verbs). Importantly, objects were categorized into two conceptual classes, i.e. animate vs inanimate objects; and two perceptual classes, i.e. black line drawings vs coloured photographs. We pseudo-randomly drew 64 images per participant according to a fully balanced scheme, such that each of the two-by-two categories included the same

number of pictures (16 animate-photographs, 16 animate -drawings, 16 inanimate-photographs, 16 inanimate-drawings). Action verbs were randomly assigned to images in each participant, and were presented together with pictures centrally overlaid on a white background. The stimulus presentation and timing and accuracy information collection were controlled by scripts written in Matlab 2017a (www.mathworks.com) and the Psychophysics Toolbox Version 3[60–62].

For the analysis we used customized Matlab code (https://www.mathworks.com/matlabcentral/fileexchange/64980-simple-rm-mixed-anova-for-any-design [63]; https://www.mathworks.com/matlabcentral/fileexchange/6874-two-way-repeated-measures-anova [64]), G*power 3.1[58]. Figures were created using the raincloud plots Version 1.1[65,66], ColorBrewer 2.0 (from www.ColorBrewer.org by Cynthia A. Brewer, Geography, Pennsylvania State University) and colorbrewer schemes 2.0 for Matlab (https://www.mathworks.com/matlabcentral/fileexchange/34087-cbrewer-colorbrewer-schemes-for-matlab)[67] and the Inkscape 1.0.1 (https://inkscape.org/).

**Procedure overview**. In both experimental groups, participants were informed about the experimental procedure, asked to sign an informed consent form and to perform a training run. After completion of this training, participants continued to the experimental task (Fig. 1b). On day 1, participants performed eight task blocks, each including an encoding block with eight trials, a 20 s distractor task and three practice cycles, each including two times eight practice trials. Returning after 48 h, participants finished the experiment with a final test consisting of a single retrieval cycle (see below for details). Before leaving, participants completed a written cued recall test. Participants in both experimental groups had been clearly and repeatedly informed about the final recall on test day 2 before carrying out the task on test day 1. It took participants about 70 min to perform the task on day 1, and about 20 min on day 2.

**Encoding**. In each encoding block (Fig. 1b), participants were instructed to study 8 novel verb-object pairings. A fixation cross was presented to the participants for a jittered time period between 500 and 1500 ms. An action verb was then presented for 1500 ms before an object was shown for a maximum time period of 7 s. To facilitate learning, participants were instructed to form a vivid visual mental image using the verb-object pairing. Once they had formed a strong mental image, participants were asked to press the up-arrow key, which moved the presentation on to the next trial. In the repeated retrieval group, it took participants 4.65 s on average, and in the restudy group it took them 4.34 s to proceed to the next trial ($SD_{retrieval}$ = 1.77; $SD_{restudy}$ = 1.65).

**Distractor**. After each encoding block, participants performed a self-paced distractor task for 20 s, indicating as fast as possible whether each of the consecutively presented numbers on the screen was odd or even, using a left/right key press. Feedback on the percentage of correct responses was provided at the end of each distractor phase.

**Practice**

*Repeated retrieval group*. The retrieval trials started with the presentation of a fixation cross, jittered between 500 and 1500 ms, and followed by the conceptual (animate/inanimate) or perceptual (photo/drawing) question that was displayed for 3 s, enabling participants to mentally prepare to recall the respective feature of the object that was relevant on a given trial. The verb was then displayed above the response alternatives (e.g., animate/inanimate), and participants had to retrieve the associated object and answer the question as fast as possible. Verb and question were displayed for a maximum period of 10 s or until the participant selected a response to the question. The questions were answered with left, downward and right-arrow keys.

*Restudy group*. In the restudy group, the paradigm was kept as similar to the repeated retrieval group as possible, including an attempt to equate average exposure times during practice (for which reason the restudy group data were collected after the retrieval group). The restudy trial was initiated with a fixation cross with the same jitter (500–1500 ms) as in the retrieval group, and followed by the conceptual or perceptual question that was displayed for 3 s. The verb cue and object then appeared together above the question. Again, participants were asked to use the 3 s period to prepare mentally to answer the question. When the object appeared, participants were instructed to first answer the question about the object they saw on the screen as fast as possible, and then use the remaining time to restudy the verb-object pair. In order to equate exposure times between the two groups, we set the trial duration of each of the three restudy cycles to the average response time of each of the three individual retrieval cycles from the previously collected retrieval group (cycle 1: 2.2 s, cycle 2: 1.9 s, cycle 3: 1.8 s).

**Retrieval and restudy blocks setup**. Participants of both groups completed three consecutive practice cycles, in each of which they practiced all eight verb-object associations they had learned in the previous encoding block twice, once answering a conceptual and once answering a perceptual question. This sums up to six practice trials per learned association, three with each question type. The order of

the conceptual and perceptual questions within cycles was counterbalanced as follows: In each of the three cycles, one half of the stimuli was first probed with a conceptual question and the other half with a perceptual question first. In addition, we controlled that each of the eight question-order possibilities occurred equally often for each object type (i.e., animate-photo, animate-drawing, inanimate-photo, inanimate-drawing). The percentage of correct trials was provided after the third practice cycle.

**Final test**. After 48 h, participants were asked to complete a final test, in which they performed one cued recall block with the same procedural set-up as on day 1 in the retrieval group. Participants were presented with a conceptual/perceptual probe, and asked to answer this question as fast as possible when cued with a verb. Each object was recalled once with each question type. Here, half of the stimuli was first probed with a conceptual question and the other half with a perceptual question, randomized independently with respect to the first testing day. Finally, participants were given a paper sheet, displaying all 64 action verbs, next to which they were asked to write down a verbal description of the associated object.

**Data preparation**. During data preparation, all RTs faster than 200 ms were excluded from the study. For the main analyses, RTs of correct trials were averaged and the standard deviation was calculated for both conceptual and perceptual questions, separately for the retrieval and the restudy group, and separately for the trials of each individual practice cycle per subject. Trials exceeding the average RT of a given cycle by more than three times the standard deviation were excluded in further RT analyses[25]. In the repeated retrieval group, 98.16% of the data remained after trimming the RTs of correct responses, whereas in the restudy group, 99.60% remained for our main analyses. Testing for a relationship between day 2 RTs and sheet responses, the RTs we used included correct, incorrect, and "don't remember" button press responses and trials exceeding the average RT of the given cycle by more than three times the standard deviation were excluded after the categorization of RTs.

To prepare the accuracy data, trials with responses faster than 200 ms, and objects with a missing response for either of both questions on one cycle were excluded in the related cycle. After this accuracy trimming, 99.39% of the repeated retrieval data and 93.26% of the restudy data remained. The RT data prepared for our main hypotheses met the normality assumptions.

**Analysis**. To assess our main hypotheses of interest, including RT differences over time and between groups, we performed repeated measures (rm) ANOVAs on the RTs. Testing for a semanticisation over time, we included cycles (cycle 3, cycle 4) and question type (perceptual, conceptual) as within-subjects factors. Control analyses were added that used the averaged day 1 data instead of cycle 3, such that the relevant factors were day (day 1, day 2) and question type (perceptual, conceptual). Exploring the RT gap between groups on day 2, we used question type (perceptual, conceptual) as within-subjects factor and group (retrieval, restudy) as between-subjects factor. To replicate a reversed stream, we again used an rmANOVA with question type as within-, and group as between-subjects factor. Additional rmANOVAs were carried out on accuracies to test for dependency between two features. Here, we used cycles and question types as within-subject factors. For posthoc analyses, we performed two-sided $T$-tests. Two-sided $T$-tests were also used on sheet response accuracies, to demonstrate a testing effect. Finally, two-sided $T$-tests were performed on RT gaps and task accuracies categorized according to sheet accuracies.

**Statistics and reproducibility**. Our methods and statistical results, especially the retrieval group results from day 1, reproduce and extend findings from one other study by Linde-Domingo et al.[25].

**Reporting summary**. Further information on research design is available in the Nature Research Reporting Summary linked to this article.

## Data availability
The retrieval and restudy data files that support the findings of this study can be downloaded from the "Retrieval_group" and "Restudy_group" folders respectively, hosted on the Open Science Framework under the identifier https://doi.org/10.17605/OSF.IO/WP4FU [68]. Stimulus material can be found in the BOSS database (https://sites.google.com/site/bosstimuli/home [59]). Source data are provided with this paper.

## Code availability
The custom code used in this study is available on the Open Science Framework with the identifier https://doi.org/10.17605/OSF.IO/WP4FU [68]. Source data are provided with this paper.

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

## Acknowledgements

We thank Jessica Davies, Brittany Lowe, Anya Mallerman-Bristow and Naomi Maurice for helping with data collection, and Marije ter Wal for useful analytical advice. This work was supported by a European Research Council (ERC) Starting Grant StG-2016-715714 awarded to M.W., by a project grant from the Economic and Social Sciences Research Council UK (ES/M001644/1) awarded to M.W., and a scholarship from the Midlands Integrative Biosciences Training Partnership (MIBTP) awarded to J.L.D.

## Author contributions

J.L. and M.W. designed the experiments. J.L. conducted the experiments. J.L. analysed the data. J.L., J.L.D. and M.W. contributed to the analysis approach. J.L. and M.W. wrote the first version of the manuscript and all authors contributed to reviewing and editing.

## Competing interests

The authors declare no competing interests.
