## [Peer Review File · Nature Communications]

Reviewers' Comments:

Reviewer #1:

Remarks to the Author:

Lifanov et al report the interesting and theoretically important finding that repeated recall leads to more semanticization of associative memories(verb-object pairs) over a two-day delay than does repeated study. They used reaction times to conceptual and perceptual information to test the nature of the memory representation. The RT advantage of conceptual over perceptual decisions increased with delay for repeated recall but not for repeated study. In addition, if participants only recalled one of the two features, they were more likely to remember the conceptual but not the perceptual feature than vice versa, a difference that increased over the delay period. They concluded that their findings support a hierarchically dependent model of remembering.

The paper is written and argued very well, the analyses are appropriate, and their interpretation properly balanced. I do, however, have a few concerns which I would like to see addressed.

Major comments

1. The contrast between conceptual and perceptual features does not take into account that the perceptual feature decision draw also on episodic memory to a greater extent than decisions about conceptual aspects of the item. Thus, if you remember that the to-be-recalled item is "dog" you immediately know that it refers to a living creature. Determining whether it was presented as a drawing or photograph requires that you also revive the episodic aspect. So the contrast could just as easily be between semantic memory and episodic memory.

One possibility is to see ask participants to make perceptual and conceptual decisions that both rely on prior knowledge. For example, participants can be asked whether the target item is larger or smaller than a shoe-box, or whether it is more angular or curved. Likewise, the study could be redone so that the conceptual and perceptual decisions both depend on episodic memory by having them study pairs in which the items are either related to one another or not (conceptual decision) or whether they were both presented as pictures or drawings, or whether they were different.

I don't expect the authors conduct more experiments, especially in these Covid times. I would, however, like them to consider this interpretation among the others they offer when discussing their findings.

2. The results of the RT and accuracy analyses of conceptual and perceptual decisions are conducted independently of performance on cued recall tests. Why not relate one set of analyses to the other? For example, is there any relation between the items participants recalled correctly and their performance on the conceptual and perceptual tasks, both on accuracy and RT. If correct recall depends on recollection, it may be the case that for correctly recalled items, there is a much smaller difference between conceptual and perceptual RTs than for incorrectly recalled items. If such analyses were already conducted, they escaped my notice.

3. Were participants in the restudy group aware that their memory would be tested as they were restudying the material? If they were not, perhaps they did not exert the effort to encode the material properly.

Minor comments

Line 46. to recall them frequently

Line 58. practice techniques such as

Line 86. Add reference to Sekeres et al, Neuroscience Letters, 2018.

Reviewer #2:

Remarks to the Author:

Feature-specific reaction times reveal a semanticisation of memories over time and with repeated remembering

By Lifanov et al.

This study by Lifanov and colleagues built on their prior work on the perceptual vs. conceptual coding of episodic memories. Here they used an encoding + practice + test paradigm to show the "semanticisation" of episodic memories such that, after a 48h delay, response times were faster to test questions about the conceptual properties ("animate" vs. "inanimate"?) of a set of word/picture paired associates than to test questions of perceptual properties ("photo" or "drawing"?). This was specifically the case when the practice type was retrieval practice, rather than restudy, consistent with the well-established testing effect. In addition, the test accuracy diverged for conceptual and perceptual tests after the 48h delay, such that there was a larger proportion of "conceptual only" correct responses than "perceptual only" responses, which is interpreted as consistent with semanticisation and the reversal of the perceptual-then-conceptual ordering from encoding.

Overall, I found this paper to be quite interesting and convincing. It is a modification of a prior paradigm that has been used by the authors, and so there were fairly specific predictions that were borne out by the data. There was a clever use of simple RTs and accuracies to demonstrate results consistent with a semanticisation process. I have no major concerns and only a few minor ones that I would like the authors to address.

Minor concerns

1) SAMPLE SIZES. There was a large imbalance in sample sizes between the retrieval group (n=49) and the restudy group (n=24). What the 2x difference? I would like the authors to verify the results by randomly sampling n=24 participants from the full n=49 sample of the retrieval group to verify that the comparisons between groups holds.

2) RAW VALUES. For Fig. 3, the difference in perceptual and conceptual RTs is shown. While this data reduction is helpful for quick interpretation, it would be valuable to see the raw scores for each condition, either in the text, table, or figure form. Were these effects driven by conceptualization becoming faster or perceptual questions being answered slower?

3) WHY ONLY CYCLE 3? In Fig. 4, the comparison is made between cycle 3 (the last testing cycle on day 1) and cycle 4 (the testing session on day 2). It is not clear why only cycle 3 was used to represent day 1, rather than say the average of cycles 1-3. Better justification is needed for this.

Reviewer #3:

Remarks to the Author:

This is an extremely well written paper with a clever experimental paradigm, presenting highly interesting results. I only have one major comment relating to the presentation of the scientific background in the manuscript and the discussion of results in light of relevant work that should be considered. This is why, in my review, I will additionally emphasize the relevance of the current findings, to back my support for publication of the results.

Rapid systems consolidation over rehearsal (or retrieval specifically) is an idea that has gained attention during the last five years. Next to theoretical accounts put forward by authors of the current

paper (Antony et al., 2016; Ferreira et al., 2019), other groups have provided imaging evidence that neocortical memory representations are formed rapidly over multiple encoding-retrieval repetitions (Brodt et al., 2016; Brodt et al., 2018; Himmer et al., 2019). These findings have been surprising against a backdrop of evidence of slow hippocampal-neocortical consolidation, and the exact mechanisms and boundary conditions of rapid systems consolidation via rehearsal are not yet well understood. Particularly behavioral evidence for systems consolidation is so far scarce, and identification of behavioral markers that may reflect the neural reorganization of memory over time is of great importance. The study of Lifanov and colleagues contributes to this quest, and helps clarifying how retrieval, in particular, may aid memory consolidation.

Major comments:

1) The introduction is very well written. I feel that the neuroimaging evidence of systems consolidation observed over repeated study and retrieval by Brodt et al., 2016; Brodt et al., 2018; Himmer et al., 2019, however, is highly relevant to the current paper, and thus think that this should be mentioned in the context of previous findings supporting the theoretical models presented.

The behavioral effects reported in this manuscript appear robust, and I consider all statistical analyses to be appropriate.

The discussion presents strong behavioral and neuroimaging evidence relating to the current findings, and the conclusions drawn by the authors are generally well supported.

2) In one particular regard, the authors' findings diverge from theoretical models (Antony et al., 2016) and previous findings (Brodt et al., 2016; Brodt et al., 2018; Himmer et al., 2019). A more in-depth discussion might benefit the scientific community here.

2.1) The authors report changes in the memory representation (indexed by the perceptual-conceptual gap) only over a longer delay of 48 hours. While this may be because time or sleep are required for this change, it may also be an issue of the sensitivity of the behavioral test, particularly if encoding-retrieval repetitions on day 1 occur without any temporal delay (and the memories will thus generally be very strong during the first 3 tests). I thus think that while the authors' results clearly show that retrieval practice has an advantage over mere restudy, and that semantic features benefit more from retrieval practice than perceptual features do, strong conclusions over the temporal dynamics of this effect, or the contribution of sleep, are premature. Conclusions about the temporal dynamics would require an appropriate control condition. Retrieval repetitions could be differentially spaced in time, or items would need to be practiced a different number of times to tease these effects apart.

2.2) A number of studies suggest that in other experimental designs, systems consolidation may be achieved within a single learning session: Using brain imaging, Brodt et al., 2016, 2018, show functional and structural plastic changes in the neocortex when participants repeatedly studied and retrieved the content. Though they did not test for behavioral changes that may index qualitative changes of the memory, these results should not be immediately discounted. It should be noted that Himmer et al., 2019 show similar changes in functional brain activity as Brodt et al., 2016, 2018, yet these changes only lasted if sleep followed learning, in line with the findings presented in the current manuscript.

2.3) I agree that plastic changes in the neocortex may not always equal stable and mature memory engrams, and potentially the resulting qualitative changes in the memory representations may need more time to surface, as suggested by the authors' results. A more parsimonious explanation, however, could be that the time and repetitions needed to form independent neocortical representations may depend on the complexity of the learning task, the retrieval demands, and similar meta-mnemonic features. I would thus appreciate if the discrepant results could be discussed. I think discussion of all the available literature that speaks to the theoretical account of Antony et al., 2016

will accelerate scientific progress on delaminating boundary conditions on when and how retrieval practice or restudy can give rise to systems memory consolidation.

Altogether, this manuscript has been a joy to read and provides highly relevant findings for everyone interested in systems consolidation and how memory representations evolve over repeated learning and time. I thus strongly encourage publication of these findings!

Additional minor remarks:

3) I would have appreciated a one sentence summary of the meaning of the findings presented in the graphics in the figure legends.

4) Whereas a small number of studies of sleep and memory suggest that categorical information shows preferential consolidation during sleep, the overwhelming majority of studies on the effect of sleep on memory actually show more modest effects, in line with a mere strengthening of learnt information. Given that theoretical accounts have long postulated qualitative changes of memories over sleep, this lack of evidence is not owing to lack of studies conducted. I thus advise caution when interpreting such findings, they may depend on task demands (i.e. is abstraction or detail required for optimal performance).

Side note: The behavioral measure of systems consolidation the authors report here, lends itself well to testing for qualitative changes of sleep on memory! The current study did not manipulate the state of consciousness explicitly, such that further studies will be needed to draw stronger conclusions from the pattern of results presented here. The results are, however, intriguing!

5) The authors state that most studies on sleep and memory employ a retrieval test before participants go to bed. This is correct. They speculate that this may be necessary to observe effects of sleep on memory performance. A group in Germany has explicitly manipulated whether participants retrieve or restudy information, and how this relates to benefits of sleep on memory consolidation (notably in a very large experimental sample). Contrary to the authors' speculation, they find that retrieval practice diminishes effects of sleep on memory (Bäuml et al., 2014). Given that other groups do report sleep effects, even if memories are retrieved beforehand, I suggest that rather than retrieval or no, the level of performance pre-sleep/strength of the memory tested may be critical to whether we can detect effects of sleep on memory performance. In other words: the sensitivity of our behavioral tests, and their timing, may matter (e.g. see Petzka et al.).

Reviewer #1 (Remarks to the Author):

Lifanov et al report the interesting and theoretically important finding that repeated recall leads to more semanticisation of associative memories(verb-object pairs) over a two-day delay than does repeated study. They used reaction times to conceptual and perceptual information to test the nature of the memory representation. The RT advantage of conceptual over perceptual decisions increased with delay for repeated recall but not for repeated study. In addition, if participants only recalled one of the two features, they were more likely to remember the conceptual but not the perceptual feature than vice versa, a difference that increased over the delay period. They concluded that their findings support a hierarchically dependent model of remembering.

The paper is written and argued very well, the analyses are appropriate, and their interpretation properly balanced. I do, however, have a few concerns which I would like to see addressed.

Major comments

1. The contrast between conceptual and perceptual features does not take into account that the perceptual feature decision draw also on episodic memory to a greater extent than decisions about conceptual aspects of the item. Thus, if you remember that the to-be-recalled item is “dog” you immediately know that it refers to a living creature. Determining whether it was presented as a drawing or photograph requires that you also revive the episodic aspect. So the contrast could just as easily be between semantic memory and episodic memory.

One possibility is to see ask participants to make perceptual and conceptual decisions that both rely on prior knowledge. For example, participants can be asked whether the target item is larger or smaller than a shoe-box, or whether it is more angular or curved. Likewise, the study could be redone so that the conceptual and perceptual decisions both depend on episodic memory by having them study pairs in which the items are either related to one another or not (conceptual decision) or whether they were both presented as pictures or drawings, or whether they were different.

I don't expect the authors conduct more experiments, especially in these Covid times. I would, however, like them to consider this interpretation among the others they offer when discussing their findings.

The reviewer raises an important and very interesting discussion point here. Throughout the paper (and in our previous work), we work under the framework that when a visual image is reconstructed from memory, an episodic trace first needs to be accessed in the hippocampus, followed by a backpropagation of information that progresses along a semantic-to-perceptual pathway. We agree with the reviewer that our findings could in principle also reflect other gradients (as we had already indicated in Ill. 53-56), and we now dedicate a paragraph of the discussion to such alternative interpretations (Il. 480-495). For example, as correctly pointed out, answering the perceptual questions requires more vivid and detailed recollection than answering the conceptual questions. It is thus well possible that the “reverse stream” reflects a semantic-to-episodic, rather than a semantic-to-perceptual gradient.

This is particularly relevant because semantic features (e.g. that a dog is animate) are inherent in an object's identity, whereas the perceptual features as used in our study are random bindings, and should thus engage episodic memory to a greater extent. There are certainly cases where the

perceptual information (e.g., colour) is part of an item's semantic representation, i.e., it is naturally bound to an item's identity (e.g., bananas are yellow). In such cases, we would in fact expect to observe that such features are reconstructed early on during memory recall.

Having said that, we were intrigued by the question raised by the reviewer, and conducted additional analyses to check whether within participants, those items for which the episodic memory is strong enough to support cued recall on the final test (day 2 paper & pencil) show a smaller RT gap. We find the opposite pattern, where accurate cued recall is associated with a larger perceptual-conceptual gap than inaccurate recall, indicating that good episodic memory is associated with fast access to semantic features. The corresponding results are explained below, under point 2, and in ll. 314-348 of the revised manuscript.

2. The results of the RT and accuracy analyses of conceptual and perceptual decisions are conducted independently of performance on cued recall tests. Why not relate one set of analyses to the other? For example, is there any relation between the items participants recalled correctly and their performance on the conceptual and perceptual tasks, both on accuracy and RT. If correct recall depends on recollection, it may be the case that for correctly recalled items, there is a much smaller difference between conceptual and perceptual RTs than for incorrectly recalled items. If such analyses were already conducted, they escaped my notice.

We thank the reviewer for this excellent suggestion, and added a new analysis in our manuscript. All reaction times have been categorized into responses where the object has been remembered on the response sheets (specific/coarse correct) or where it has not been remembered (specific/coarse incorrect). Results show that the perceptual-conceptual RT gap is in fact significantly larger for correctly than for incorrectly recalled items ($t = 2.13$, $p = .04$ with the specific scoring approach, $t = 2.65$, $p = .01$ with the coarse scoring approach). This result suggests that those participants with good episodic memory tend to show a larger gap, rather than a smaller one, consistent with semanticisation aiding episodic recall on delayed tests. Note that a mirrored effect was found for accuracies, such that participants who performed better on the paper-and-pencil test, showed a larger accuracy gap in the button presses according to the coarse ($T(71) = 3.95$, $p < .01$, $CI = [.05, .16]$) and specific ($T(71) = 3.08$, $p < .01$, $CI = [.03, .15]$) scoring method. The new results are reported in ll. 314-348 of the revised manuscript, and are discussed in ll. 421-426 of the discussion section.

3. Were participants in the restudy group aware that their memory would be tested as they were restudying the material? If they were not, perhaps they did not exert the effort to encode the material properly.

We thank the reviewer for pointing out this potential concern. Participants in both groups were clearly and repeatedly informed of the fact that the practiced associations would be tested when returning to the lab after two days, and that they should thus make an effort to learn the associations as well as possible. We added a corresponding statement in our manuscript under the section "Procedure overview" (ll. 568-569).

Minor comments

Line 46. to recall them frequently

Line 58. practice techniques such as

Line 86. Add reference to Sekeres et al, Neuroscience Letters, 2018.

We thank the reviewer for these hints and made the appropriate changes to the manuscript.

Reviewer #2 (Remarks to the Author):

Feature-specific reaction times reveal a semanticisation of memories over time and with repeated remembering

By Lifanov et al.

This study by Lifanov and colleagues built on their prior work on the perceptual vs. conceptual coding of episodic memories. Here they used an encoding + practice + test paradigm to show the “semanticisation” of episodic memories such that, after a 48h delay, response times were faster to test questions about the conceptual properties (“animate” vs. “inanimate”?) of a set of word/picture paired associates than to test questions of perceptual properties (“photo” or “drawing”?). This was specifically the case when the practice type was retrieval practice, rather than restudy, consistent with the well-established testing effect. In addition, the test accuracy diverged for conceptual and perceptual tests after the 48h delay, such that there was a larger proportion of “conceptual only” correct responses than “perceptual only” responses, which is interpreted as consistent with semanticisation and the reversal of the perceptual-then-conceptual ordering from encoding.

Overall, I found this paper to be quite interesting and convincing. It is a modification of a prior paradigm that has been used by the authors, and so there were fairly specific predictions that were borne out by the data. There was a clever use of simple RTs and accuracies to demonstrate results consistent with a semanticisation process. I have no major concerns and only a few minor ones that I would like the authors to address.

Minor concerns

1) SAMPLE SIZES. There was a large imbalance in sample sizes between the retrieval group (n=49) and the restudy group (n=24). What the 2x difference? I would like the authors to verify the results by randomly sampling n=24 participants from the full n=49 sample of the retrieval group to verify that the comparisons between groups holds.

We understand the reviewer’s concern about the different sample sizes. We thus added a random subsampling analysis to further underpin the difference of the perceptual-conceptual gap between the restudy and the repeated retrieval group. We randomly drew 5000 subsamples from the retrieval group, and compared the difference (z-score) between the perceptual-conceptual RT gap in the restudy group (original sample of n = 24) with the distribution of resampled retrieval groups (each subsampled randomly at n = 24). We found that a) the mean RT gaps in the subsampled retrieval groups (n=24) distributed around the mean RT gap observed for the larger sample (n=49), with a 95% confidence interval of 203 to 376 msec; b) the between group comparison based on subsampling yielded similar results as our old methods, resulting in a highly significant z-score (z=-3.95, p<.01) and empirical p-value (p<.01) (i.e. in zero out of 5000 subsamples was the retrieval gap smaller than the restudy gap). This result confirms that the stronger semanticisation in the repeated retrieval than

restudy group also holds with equal sample sizes, and has been added to the Supplementary Methods (see Supplementary Figure 1a)

The same subsampling analysis has been added to the section about the testing effect. We found that the restudy accuracy significantly differed from the repeated retrieval accuracy distribution for both coarse (z -score=-3.63, $p<.01$, empirical $p<.01$) and specific accuracies (z -score=-3.45, $p<.01$, empirical $p<.01$ s). This finding confirms that our data, not surprisingly, did show a testing effect in commonly used cued recall accuracies (Supplementary Methods and Supplementary Figure 1b-c)

2) RAW VALUES. For Fig. 3, the difference in perceptual and conceptual RTs is shown. While this data reduction is helpful for quick interpretation, it would be valuable to see the raw scores for each condition, either in the text, table, or figure form. Were these effects driven by conceptualization becoming faster or perceptual questions being answered slower?

We agree that the raw values are important for interpreting the results, and thus added the means and standard deviations in the manuscript. As can be seen, perceptual questions are answered only slightly faster in the retrieval group in comparison to the restudy group. In contrast the difference between the conceptual RT differs comparatively more between repeated retrieval and restudy. These raw values suggest that the reported interaction is mainly driven by a stronger conceptualization within the repeated retrieval group, rather than a change in perceptual RTs (ll. 222-230), in line with our interpretation.

3) WHY ONLY CYCLE 3? In Fig. 4, the comparison is made between cycle 3 (the last testing cycle on day 1) and cycle 4 (the testing session on day 2). It is not clear why only cycle 3 was used to represent day 1, rather than say the average of cycles 1-3. Better justification is needed for this.

We assume that Reviewer 2 is here referring to the first main hypothesis, which tests for the semanticisation over time in the repeated retrieval group. The rationale for this comparison was to isolate the transformation of episodic memories that occurs purely with passage of time from the changes that occur already on the first day, across the repeated practice trials. We thus aimed at probing the memory representation as formed after completed learning and practice on day 1 (that is, the “end state” of the memory on day 1), and compare it to the changed representation on day 2. Note that while our data do not support this suggestion, in theory it would have been possible that repeated remembering semanticises memories already on day 1, in line with a “fast consolidation” (see Antony et al., 2017; Brodt et al., 2016, 2018). If so, the perceptual-conceptual gap would have increased from cycle 1 to cycle 3, and it would have been even more important to compare cycle 3 (end state of day 1) to cycle 4 (state on day 2) in order to isolate the additional semanticisation over time. The same logic applies to the analysis on the hierarchical relationship between remembered features.

Having said that, to reassure the reviewer, we performed the same analyses comparing the day 2 data to the averaged data of all three cycles on day 1, as suggested. The repeated measures ANOVA with the factors time (day 1 vs day 2), where the day 1 data included the within-subject average responses over the three cycles of day one for both questions individually, and feature (conceptual vs perceptual) confirms a significant effect of time ($F(1,48)=43.97$, $p<.01$), a main effect of question type ($F(1,48)=29.92$, $p<.01$) and most importantly, a significant interaction ($F(1,48)=20.11$, $p<.01$; ll. 183-185).

We also checked if the dependency analyses held true after averaging responses of the three cycles on day 1, and we can confirm the increasing hierarchical dependence of remembering on conceptual features, with a significant effect of time (day 1 vs day 2) ($F(1,48)=64.25, p<.01$), a main effect of feature remembered ($F(1,48)=29.15, p<.01$), and the expected significant interaction ($F(1,48)=14.31, p<.01$; ll. 282-284).

As the interaction effects in the two above analyses are our main effect of interest, we have shortly mentioned, that the interaction effects remain robust after averaging data over day1 in the relevant sections (ll. 183-185; 282-284). Moreover, we added a justification of our chosen analysis approach to the main text (p. ll. 165-168).

Reviewer #3 (Remarks to the Author):

This is an extremely well written paper with a clever experimental paradigm, presenting highly interesting results. I only have one major comment relating to the presentation of the scientific background in the manuscript and the discussion of results in light of relevant work that should be considered. This is why, in my review, I will additionally emphasize the relevance of the current findings, to back my support for publication of the results.

Rapid systems consolidation over rehearsal (or retrieval specifically) is an idea that has gained attention during the last five years. Next to theoretical accounts put forward by authors of the current paper (Antony et al., 2016; Ferreira et al. 2019), other groups have provided imaging evidence that neocortical memory representations are formed rapidly over multiple encoding-retrieval repetitions (Brodt et al., 2016; Brodt et al., 2018; Himmer et al., 2019). These findings have been surprising against a backdrop of evidence of slow hippocampal-neocortical consolidation, and the exact mechanisms and boundary conditions of rapid systems consolidation via rehearsal are not yet well understood. Particularly behavioral evidence for systems consolidation is so far scarce, and identification of behavioral markers that may reflect the neural reorganization of memory over time is of great importance. The study of Lifanov and colleagues contributes to this quest, and helps clarifying how retrieval, in particular, may aid memory consolidation.

Major comments:

1) The introduction is very well written. I feel that the neuroimaging evidence of systems consolidation observed over repeated study and retrieval by Brodt et al., 2016; Brodt et al., 2018; Himmer et al., 2019, however, is highly relevant to the current paper, and thus think that this should be mentioned in the context of previous findings supporting the theoretical models presented.

The behavioral effects reported in this manuscript appear robust, and I consider all statistical analyses to be appropriate.

The discussion presents strong behavioral and neuroimaging evidence relating to the current findings, and the conclusions drawn by the authors are generally well supported.

We sincerely thank the reviewer for this positive feedback. We agree that the fast consolidation work mentioned by the reviewer is highly relevant for putting our work in context, and we took the additional literature into account for our revisions. The relevant papers are now cited in the introduction (ll. 90-93), and are discussed in some detail in a new discussion paragraph (ll. 451-463).

2) In one particular regard, the authors' findings diverge from theoretical models (Antony et al., 2016) and previous findings (Brodt et al., 2016; Brodt et al., 2018; Himmer et al., 2019). A more in-depth discussion might benefit the scientific community here.

2.1) The authors report changes in the memory representation (indexed by the perceptual-conceptual gap) only over a longer delay of 48 hours. While this may be because time or sleep are required for this change, it may also be an issue of the sensitivity of the behavioral test, particularly if encoding-retrieval repetitions on day 1 occur without any temporal delay (and the memories will thus generally be very strong during the first 3 tests). I thus think that while the authors' results clearly show that retrieval practice has an advantage over mere restudy, and that semantic features benefit more from retrieval practice than perceptual features do, strong conclusions over the temporal dynamics of this effect, or the contribution of sleep, are premature. Conclusions about the temporal dynamics would require an appropriate control condition. Retrieval repetitions could be differentially spaced in time, or items would need to be practiced a different number of times to tease these effects apart.

We appreciate the diverging results provided by (Brodt et al., 2016; Brodt et al., 2018; Himmer et al., 2019) and now discuss these in our revised discussion (ll. 451-463). Specifically, we agree that it would be very interesting to see how fast memory representations change (e.g., do we need 3 or 20 repetitions to induce a lasting change), and if breaks are required for a behavioural change to emerge. The studies referred to here all used a larger number, and more spaced, repetitions of the to-be-learned information, and we might have observed semanticisation with similar numbers of repetitions and spaced intervals in our study. These are now discussed in ll. 458-460. We also agree that the specific influence of sleep cannot be established in our design, and have rephrased the discussion accordingly (ll.430-435) (also see our answer to point 2.4 below).

2.2) A number of studies suggest that in other experimental designs, systems consolidation may be achieved within a single learning session: Using brain imaging, Brodt et al., 2016, 2018, show functional and structural plastic changes in the neocortex when participants repeatedly studied and retrieved the content. Though they did not test for behavioral changes that may index qualitative changes of the memory, these results should not be immediately discounted. It should be noted that Himmer et al., 2019 show similar changes in functional brain activity as Brodt et al., 2016, 2018, yet these changes only lasted if sleep followed learning, in line with the findings presented in the current manuscript.

We thank the reviewer for this remark and added a more in-depth discussion of this apparent discrepancy in the literature (ll. 451-463). We discuss neuroimaging evidence, where indeed some studies (Himmer et al., 2019; Ferreira et al., 2019) suggest that on a neural level, rehearsal effects require a period of post-practice consolidation to take effect; and other studies (Brodt et al., 2016, 2018) suggesting that structural and functional changes in the neocortex develop within a single session. The reviewer points out that the latter findings contradict our behavioural null results on day 1, suggesting no qualitative change in memories in the first session. We would argue, in line with the reviewer, that there is no contradiction, and these seemingly conflicting findings can be explained within the same framework. Considering the between-group differences on day 2 observed in our study, the fact that semanticisation was stronger in the repeated retrieval group on day 2 implies that there must have been some diverging brain processes during repeated retrieval that were at least partially responsible for the semanticisation, in line with, Brodt et al., 2016, 2018. The remaining question then is, why we cannot see the behavioural changes on a more rapid timescale, when the neurobiological changes presumably take place. In our view, the most likely explanation is

that after initial learning + practice, complementary memory traces co-exist in hippocampus and neocortex (with a stronger neocortical trace after retrieval practice), and both can support successful remembering. As pointed out by the reviewer, the episodic (i.e., hippocampal) trace will be very strong and dominates recall briefly after learning + practice. After a 2-day delay, the hippocampal trace might be harder to access, and recall thus depend more on neocortex (e.g., parietal lobe as in Brodt et al., 2016, 2018; Himmer et al., 2019; Ferreira et al., 2019). We would therefore argue that initial retrieval practice helps to establish the neocortical representation, but this might not be required for remembering until day 2, where the behavioural effect of having access to a neocortical representation becomes manifest. We had briefly mentioned this possibility in our previous version, but are now discussing in more detail how the various neural and behavioural findings can be reconciled (ll. 451-463).

2.3) I agree that plastic changes in the neocortex may not always equal stable and mature memory engrams, and potentially the resulting qualitative changes in the memory representations may need more time to surface, as suggested by the authors' results. A more parsimonious explanation, however, could be that the time and repetitions needed to form independent neocortical representations may depend on the complexity of the learning task, the retrieval demands, and similar meta-mnemonic features. I would thus appreciate if the discrepant results could be discussed. I think discussion of all the available literature that speaks to the theoretical account of Antony et al., 2016 will accelerate scientific progress on delimiting boundary conditions on when and how retrieval practice or restudy can give rise to systems memory consolidation.

Altogether, this manuscript has been a joy to read and provides highly relevant findings for everyone interested in systems consolidation and how memory representations evolve over repeated learning and time. I thus strongly encourage publication of these findings!

We thank the reviewer for this very positive assessment, and distilling our findings so comprehensively in light of the relevant literature.

Additional minor remarks:

3) I would have appreciated a one sentence summary of the meaning of the findings presented in the graphics in the figure legends.

We thank the reviewer for this suggestion. Summaries have been added to the figure legends.

4) Whereas a small number of studies of sleep and memory suggest that categorical information shows preferential consolidation during sleep, the overwhelming majority of studies on the effect of sleep on memory actually show more modest effects, in line with a mere strengthening of learnt information. Given that theoretical accounts have long postulated qualitative changes of memories over sleep, this lack of evidence is not owing to lack of studies conducted. I thus advise caution when interpreting such findings, they may depend on task demands (i.e. is abstraction or detail required for optimal performance).

Side note: The behavioral measure of systems consolidation the authors report here, lends itself well to testing for qualitative changes of sleep on memory! The current study did not manipulate the

state of consciousness explicitly, such that further studies will be needed to draw stronger conclusions from the pattern of results presented here. The results are, however, intriguing!

We do take the reviewer's point on board, and are now more cautious not to over-interpret the studies that seemingly show qualitative changes with sleep and time passing. We also explicitly mention that our simple behavioural task lends itself well to studying the specific effects of sleep (ll. 467-469). We also agree that the effects of retrieval practice and consolidation will depend on task demands, and briefly discuss this point (see answer to point 5 below).

5) The authors state that most studies on sleep and memory employ a retrieval test before participants go to bed. This is correct. They speculate that this may be necessary to observe effects of sleep on memory performance. A group in Germany has explicitly manipulated whether participants retrieve or restudy information, and how this relates to benefits of sleep on memory consolidation (notably in a very large experimental sample). Contrary to the authors' speculation, they find that retrieval practice diminishes effects of sleep on memory (Bäuml et al., 2014). Given that other groups do report sleep effects, even if memories are retrieved beforehand, I suggest that rather than retrieval or no, the level of performance pre-sleep/strength of the memory tested may be critical to whether we can detect effects of sleep on memory performance. In other words: the sensitivity of our behavioral tests, and their timing, may matter (e.g. see Petzka et al.).

This is a fair point, and we never meant to suggest, in our discussion, that sleep effects cannot occur without a pre-sleep retrieval test. We merely pointed out that it is important to disentangle retrieval's specific contribution to the sleep effects that have been reported in the literature (ll. 427-435).

We also agree that the study by Bäuml et al. is in direct contradiction to the idea that retrieval boost the subsequent consolidation effects. In fact, however, the large majority of testing effect studies do show that retrieval boosts long-term retention of a memory in particular, and the authors of the Bäuml et al. (2014) paper themselves interpret their observations as reflecting non-trivial ceiling/floor effects: if a memory's strength is pushed above a certain strength threshold already before sleep, through repeated retrieval practice, an additional sleep benefit is unlikely (see Petzka et al.'s analysis for a similar argument). This relevant literature is now discussed in ll. 427-444.

Reviewers' Comments:

Reviewer #1:

Remarks to the Author:

I commend the authors on their revision. They addressed all of my comments and, I think, those of the other reviewers very well.

It's an excellent paper that should have a strong impact on the field.

Reviewer #2:

Remarks to the Author:

We are quite satisfied with the authors' rebuttals and changes to the manuscript.

Reviewer #3:

Remarks to the Author:

I thank the authors for the very careful consideration of my comments. All my suggestions have been fully addressed.

I agree with the authors that after initial practice, complementary memory traces co-exist in the hippocampus and neocortex, but the neocortical trace may take more time (or sleep) to mature, before it can be independently accessed. I appreciate that this I now discussed in the manuscript.

Finally, I want to emphasize that the additional analyses the authors have added in response to other comments further strengthen this excellent paper.

I am excited to add this work to our lab's general reading canon. Congratulations to the authors!